evolution, ecology, bioinformatics

phylogenomics, cospeciation, Columbidae, ectoparasite, lice

**Author for correspondence:**
Bret M. Boyd
e-mail: boydbm@vcu.edu

# Long-distance dispersal of pigeons and doves generated new ecological opportunities for host-switching and adaptive radiation by their parasites

Bret M. Boyd[1], Nam-Phuong Nguyen[2], Julie M. Allen[3],
Robert M. Waterhouse[4], Kyle B. Vo[1], Andrew D. Sweet[5], Dale H. Clayton[6],
Sarah E. Bush[6], Michael D. Shapiro[6] and Kevin P. Johnson[7]

[1]Center for Biological Data Science, Virginia Commonwealth University, Richmond, VA, USA
[2]Department of Computer Science, University of Illinois, Champaign, IL, USA
[3]Department of Biology, University of Nevada, Reno, Reno, NV, USA
[4]Department of Ecology and Evolution, University of Lausanne and Swiss Institute of Bioinformatics, Lausanne, Switzerland
[5]Department of Biological Sciences, Arkansas State University, Jonesboro, AR, USA
[6]School of Biological Sciences, University of Utah, Salt Lake City, UT, USA
[7]Illinois Natural History Survey, Prairie Research Institute, University of Illinois, Champaign, IL, USA

  BMB, 0000-0002-9837-3483; RMW, 0000-0003-4199-9052; ADS, 0000-0003-2765-7021;
DHC, 0000-0003-1698-3542; SEB, 0000-0002-2913-4876; MDS, 0000-0003-2900-4331;
KPJ, 0000-0002-4151-816X

Adaptive radiation is an important mechanism of organismal diversification and can be triggered by new ecological opportunities. Although poorly studied in this regard, parasites are an ideal group in which to study adaptive radiations because of their close associations with host species. Both experimental and comparative studies suggest that the ectoparasitic wing lice of pigeons and doves have adaptively radiated, leading to differences in body size and overall coloration. Here, we show that long-distance dispersal by dove hosts was central to parasite diversification because it provided new ecological opportunities for parasites to speciate after host-switching. We further show that among extant parasite lineages host-switching decreased over time, with cospeciation becoming the more dominant mode of parasite speciation. Taken together, our results suggest that host dispersal, followed by host-switching, provided novel ecological opportunities that facilitated adaptive radiation by parasites.

## 1. Introduction

Adaptive radiations have profoundly influenced the development of evolutionary theory [1,2]. An adaptive radiation occurs when speciation is triggered by new ecological opportunities [1,3]. Classic examples of adaptive radiation involve the evolution of diverse morphological features, such as bill shape in Galapagos finches [4] and Hawaiian honeycreepers [5], body shape in Caribbean anole lizards [6], mouthparts and body shape in African rift lake cichlid fishes [7], and growth habit in Hawaiian silversword plants [8]. These notable radiations occurred in isolated habitats, such as volcanic islands or recently formed lakes, in which a newly colonizing lineage diversified into multiple lineages in response to new ecological opportunities [1,3].

Relatively few studies of adaptive radiation have focused on animal parasites, despite the fact that they represent much of the world's biodiversity [9]. There is great potential for adaptive radiation by parasites, especially when a parasite lineage switches to a novel host lineage that was not previously colonized. In particular, when a parasite species colonizes a previously uninfested host lineage

(analogous to a free-living species dispersing to an uninhabited island or lake), the parasite may subsequently switch among the species in that host lineage, leading to parasite diversification [9]. This process amounts to adaptive radiation of parasites across host 'islands'.

Two recent studies identified the ecological mechanisms of adaptive speciation within a group of animal parasites [10,11], the wing lice (Insecta: Phthiraptera: *Columbicola*) of pigeons and doves (hereafter doves; Aves: Columbiformes). Wing lice are permanent ectoparasites that pass all stages of their life cycle on the body of the host, feeding on downy feathers and escaping from host preening through insertion between the feather barbs of flight feathers [9]. Most species of wing lice parasitize a single host species. In these lice, body size and colour are both correlated with the size and colour of their hosts [10,12,13]. Recent experimental studies demonstrate the adaptive basis of size and colour [10,11]. Wing lice (*Columbicola columbae*) transferred to pigeon breeds that differ dramatically in size (simulating host-switching), rapidly evolve differences in size within a few generations [11]. Size is an adaptation to escape host-preening defence [11,12]. Changes in body size trigger reproductive isolation, which is a precursor to speciation, due to a mismatch between male and female body sizes that limits copulation [11].

Through similar transfer experiments, louse coloration was shown to be an adaptation for crypsis, such that lice transferred to white pigeon breeds became lighter and those to dark breeds became darker [10]. Again, this adaptation was selected by host-preening defence, as birds more readily removed lice that did not match host coloration [10]. Together, these experiments demonstrate the adaptive basis of morphological variation across this clade of parasites and recreate the diversity of size and colour phenotypes observed across known wing louse species [10,13]. Moreover, these studies show how parasites can diversify when presented with novel ecological conditions.

Given that the adaptive mechanisms underpinning reproductive isolation have been identified through experimental work emulating host-switching [10,11], we focused on historical opportunities for host-switching that may have served as a foundation for adaptive radiations in parasites. Host dispersal and contact between host species have been recognized as having the potential to increase parasite diversity [14–16]. Here, we specifically focused on the role host dispersal plays in bringing parasite lineages into contact with novel host species with which they did not previously overlap in geographic range. We then evaluated whether this novel range overlap provided new opportunities for host-switching by identifying host dispersal events that preceded parasite speciation through host-switching [17]. If host-switching was facilitated by host sympatry, two possibilities exist: first, resident parasites may have switched to newly arriving hosts; reciprocally, parasites on the newly arriving hosts may have switched to resident hosts [18,19]. By comparing the dove and wing louse trees, we identified the direction of host-switching by parasites following host dispersal.

While ecological opportunity is essential for the occurrence of adaptive radiations, these opportunities are finite and radiating lineages will ultimately saturate open niche space [20–23]. In an adaptive radiation, the predominant mode of speciation should change over time. In host–parasite systems, parasite speciation events can be categorized as having resulted from (i) host-switching, which involves dispersal to and establishment on a host species on which a parasite did not previously occur, or (ii) cospeciation, which is the simultaneous divergence of host and parasite lineages [9]. Using cophylogenetic reconstruction and time-calibrated trees of doves and their wing lice, we classified louse speciation events as resulting from either host-switching or cospeciation. Finally, we compared the relative timing of these two events in extant lineages. Specifically, we evaluated whether the age of dove wing louse speciation events triggered by host-switching tend to be older than speciation events triggered by cospeciation. This pattern would be consistent with host lineages becoming more fully colonized by a parasite lineage over time, eventually filling up open niches and reducing ecological opportunity. Taken together, these analyses provide a framework to evaluate the phylogenetic and biogeographic underpinnings of adaptive radiation in host–parasite systems by focusing on new ecological opportunities generated by long-distance host dispersal and the relative timing of host-switching and cospeciation events.

## 2. Methods

A total of 61 species of doves were used for whole-genome sequencing. These species were selected to include dove species that are hosts to the parasitic wing lice included in a phylogenomic tree published by Boyd *et al.* [24]. This taxon sampling allowed us to infer the evolutionary history of extant lineages of doves and compare the results to evolutionary history of wing lice previously described. Genomic DNA was extracted from dove tissues and sequenced on the Illumina HiSeq platform. Protein-coding genes were identified from the *Columba livia* reference genome using the MAKER pipeline [25,26]. We then identified single-copy orthologous genes in the genome of *C. livia* and all Sauropsida available in OrthoDB v8 [27]. We then removed candidate within-clade paralogues (electronic supplementary material, methods). Sequence reads from each library were mapped to the remaining *C. livia* orthologues using Bowtie2 (v2.3.4.1) [28]. Final gene sequences for all 61 dove species were generated by taking the consensus base at each position [29] and clustered into orthologous gene sets.

We used two methods to infer the evolutionary relationships of dove species. Orthologue sets were aligned as translated amino acid sequences using UPP (v2.0) [30] and back translated to nucleotide sequences. RAxML (v8.2.9) [31] was used to estimate individual gene trees under the GTR+$\Gamma$ models and 100 bootstrap replicates. ASTRAL-II (v4.9.9) [32], a coalescence-based analysis, was used to estimate a species tree from the maximum-likelihood (ML) gene trees. Next, we sought to estimate the dove phylogeny from all of the sequence data simultaneously. We generated a concatenated alignment including all sequence data to estimate the ML tree based on this supermatrix. For each gene tree, the GTR+$\Gamma$ parameters were estimated for each codon position, and k-means clustering was applied to these codon rate parameters to create a partition for the alignment. An ML tree was estimated from the partitioned alignment using RAxML under GTR+$\Gamma$ models. A total of 100 bootstrap replicate trees were computed to infer support. Similar procedures were implemented by Boyd *et al.* [24] to infer the evolutionary relationships of wing lice, using an ML tree based on 977 single-copy orthologues (electronic supplementary material, methods).

MCMCTree, implemented in PAML (v4.9) [33], was then used to estimate the timing of dove and wing louse speciation events. We identified six dove fossils suitable as minimum age internal calibration points (electronic supplementary material, table S1 and figures S4 and S5). A maximum constraint was added to the root in order to estimate global DNA substitution

rates. Two different analyses were conducted to accommodate two different root constraints (electronic supplementary material, table S1). The oldest fossil in the tree, *Arenicolumba*, is a biogeographic anomaly (fossil described from New World, but closely allied species found in Old World) [34], so each analysis was done both with and without this fossil calibration. Likewise, wing louse species divergence times were estimated in millions of years using the ML tree described by Boyd *et al.* [24]. Nine internal calibration points were used based on terminal cospeciation events, in which terminal sister species of lice were associated with terminal sister species of hosts (electronic supplementary material, table S1, internal calibration ranges described in electronic supplementary material, figures S5–S8). The age ranges of these internal calibration points were based on the 95% confidence intervals of the estimated dates of these nodes in host trees. Maximum constraints were added to the root and to estimate global DNA substitution rates (electronic supplementary material, table S1 and figures S5–S8). We conducted four analyses in total using all possible combinations of two different maximum constraints and two different sets of internal calibration points (based on different estimates of the timing of cospeciation given the two different host trees). In both dove and louse trees, branch lengths were estimated using baseml and then divergence times were estimated under a GTR+$\Gamma$ model with Markov chain Monte Carlo (MCMC) approximation and an independent rates model. We determined that node age estimates converged between identical runs using Tracer (v1.7.1) [35]. Additionally, a time-calibrated tree was then used to estimate the ancestral species ranges of doves using BioGeoBEARS (v1.1.1) [36] under the optimal model of DIVA-like + J.

PACo was used to test whether host and parasite trees were significantly concordant, which can be a signature of cospeciation [37]. Given that we found significant congruence (see Results), we used Jane (v4) [38] to reconstruct specific cophylogenetic events, including cospeciation, host-switching, parasite duplication, parasite failure to diverge and sorting events (extinction). In total, three comparisons were conducted. First, we compared louse and host trees disregarding node ages. Second, we compared louse and host trees while binning speciation events within three different time intervals based on node age. The three intervals were based on a review of Gondwanan biotic interchange (66–45 Ma, 45–30 Ma and 30–0 Ma) [39]. Time-constrained comparisons were conducted with one louse time-tree and both available host time-trees for both Jane and PACo (additional comparisons were not needed, given the time intervals used, because the node age category assignments were identical). In all comparisons, nodes in the louse tree with less than 85% bootstrap support were collapsed to reflect ambiguity around species relationships. Using the BioGeoBEARS reconstruction, we identified lineages in the tree that were inferred to have colonized one biogeographic region from another. By combining the results from Jane and BioGeoBEARS, we evaluated whether dove lineages moving from one region to another (i) acquired lice from a dove lineage already in that region or (ii) carried lice to a new region where they switched to dove lineages already in the region. We also categorized each speciation event (i.e. node) in the louse tree as the result of (i) host-switching or (ii) cospeciation. We did not find evidence of parasite duplication events (speciation of parasite in absence of host speciation) from the Jane reconstructions. We predicted that these events would be rare, because it is unusual for a single species of dove to host more than one species of wing louse. Other cophylogenetic events, such as failure to speciate and sorting, do no result in parasite speciation and are thus not relevant to the categorization of parasite speciation events. Using the estimated dates for each of these events, we plotted the relative ratio of cospeciation and host-switching events to overall speciation events and the generation of extant parasite diversity over time.

# 3. Results

We identified 6363 single-copy orthologues from which to infer the phylogenetic relationships of 61 dove species. Simultaneous analysis of a supermatrix (11 103 960 bases) produced a well-supported tree (100% bootstrap support for 62 of 63 nodes; electronic supplementary material, figure S1) that was largely in agreement with a coalescence-based species tree (0.99–1 posterior probability for 60 of 63 nodes; electronic supplementary material, figure S2). Species divergence times were estimated using two different sets of root maximum constraints, and these analyses suggest that doves began to diversify around 51 Ma or 60 Ma depending on root age calibration (figure 1; electronic supplementary material, figures S3 and S4). Removing a biogeographically unusual and comparatively old fossil from the genus *Arenicolumba* resulted in node ages that were only slightly younger (45 Ma or 55 Ma), indicating that this fossil did not have a dramatic impact on the results. Species divergence time estimates for wing lice showed diversification began around either 15 Ma or 24 Ma depending on root age calibration (electronic supplementary material, figures S5–S8). The 95% confidence intervals for node ages for the earliest divergence in the dove tree versus louse tree did not overlap, with dove diversification starting prior to louse diversification in all comparisons (electronic supplementary material, figures S9–S14). Both dove and wing louse trees were structured by geographic region, and ancestral area reconstruction suggested that Australasia + New World was the ancestral range of doves. The earliest divergence splits the dove tree into two clades, one with an Australasian origin (Raphinae) and the other with a New World origin (Claravinae + Columbinae; figure 1). During the Neogene, members of the Australasian clade went on to colonize Eurasia and Africa, while members of the New World clade went on to colonize Australasia, Eurasia and Africa.

A comparison of calibrated host and parasite trees detected the presence of significant congruence, providing evidence of significant widespread cospeciation (PACo, residual sum of squares 9.932, $p < 0.0001$; or 15.93, $p < 0.0001$, depending on host basal calibration of 71 Ma or 89 Ma, respectively). An initial comparison of the dove and wing louse trees using an event-based method (Jane), without considering the timing of speciation events, suggests that the lice had an ancestral host among the Australasian dove clade (electronic supplementary material, table S2). When the timing of louse speciation events is constrained by time intervals informed by habitat connectivity (wing louse diversification beginning at 15 Ma or 24 Ma), the analysis again supported a single geographic origin of wing lice. However, the ancestral host of wing lice is unclear, with four possible origins: one in Australasia, two in the New World and one in Africa (electronic supplementary material, figures S15–S17 and table S2). Comparisons of dove and louse trees identified 32 cospeciation events and 28 host-switching events (figure 2; electronic supplementary material, table S2). The overall proportion of cumulative wing louse speciation events attributable to cospeciation, resulting in extant wing louse lineages and species, increased over time (figure 3; electronic supplementary material, figure S12). The median age of louse speciation events due to cospeciation was significantly younger than those due to host-switching (Wilcoxon signed-rank test, $p = 0.021$). A comparison of host and parasite trees without consideration of time constraints found additional cospeciation events (electronic supplementary

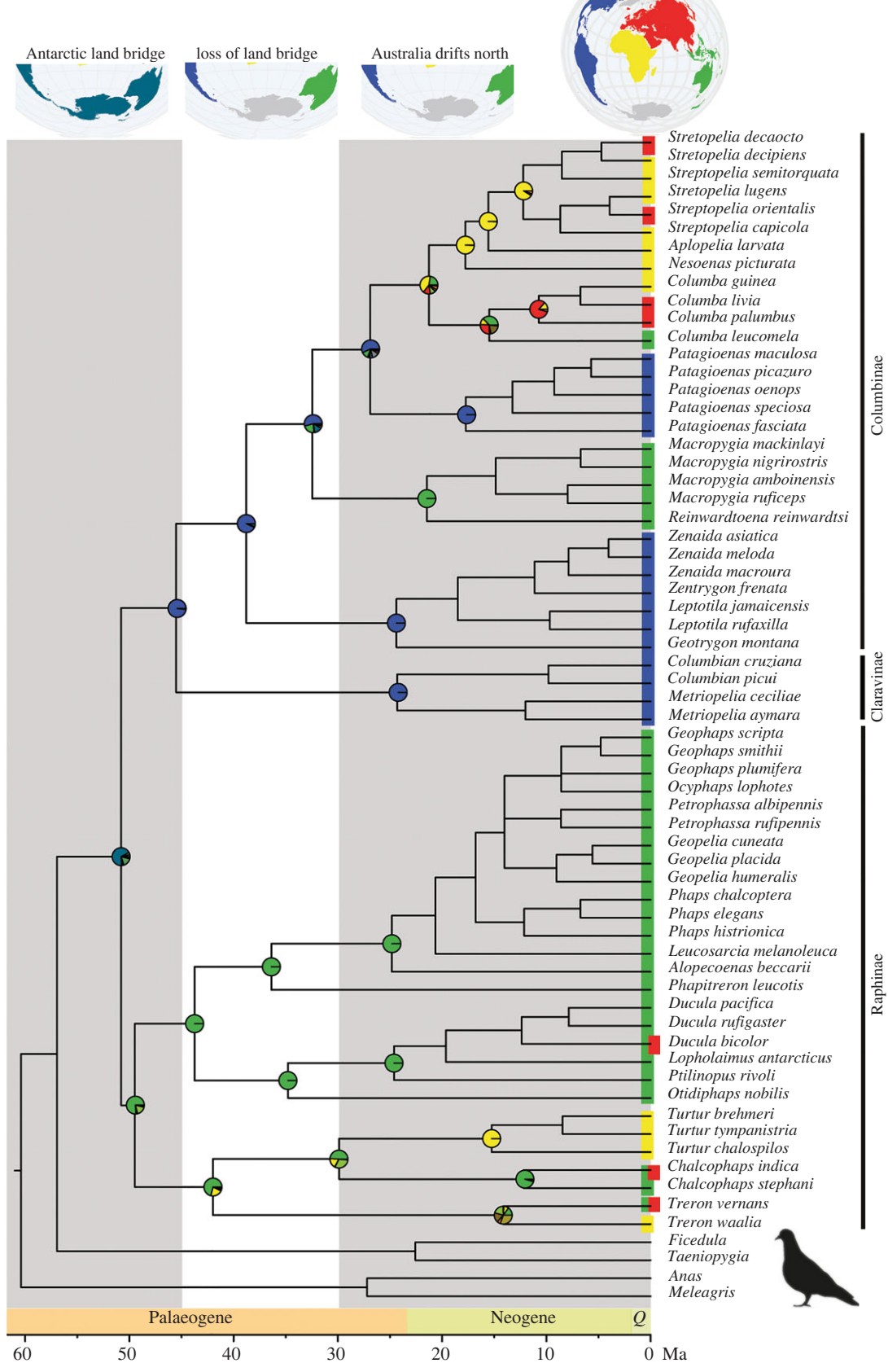

**Figure 1.** Time-calibrated phylogeny of doves based on sequences of 6363 nuclear single-copy orthologues. Scale bar at base of figure represents age in millions of years ago and geographic periods are designated above scale bar. Grey and white columns represent time intervals significant to biotic interchange across the Australia–Antarctica–South American land bridge and subsequent loss of habitat connectivity used to inform host–parasite phylogeny comparisons (66–45, 45–30, and 30–0 Ma). Colours at tree tips represent modern geographic ranges of pigeon and dove species and circles at nodes represent estimated ancestral ranges corresponding with the globe at the top of the tree. Blue-green colour in basal circle represents an ancestral range combining Australia and South America, corresponding to the Australia–Antarctica–South American land bridge illustrated at top of the figure. Q = Quaternary. Ma = millions of years ago. World maps created using rnaturalearth (https://github.com/ropensci/rnaturalearth). (Online version in colour.)

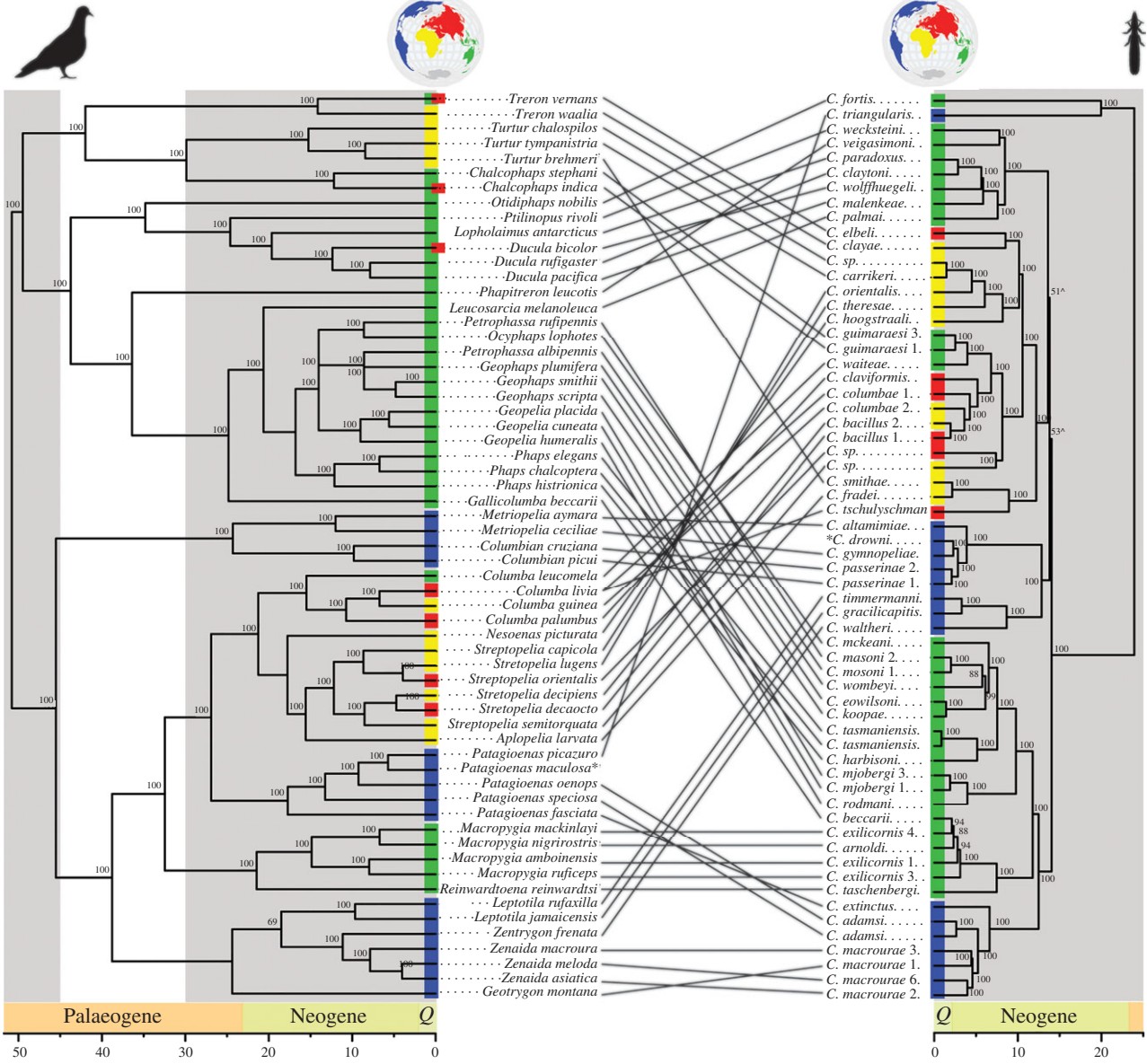

**Figure 2.** Time-calibrated phylogeny of doves based on sequences of 6363 nuclear single-copy orthologues (left) and their parasitic wing lice based on sequences of 977 single-copy orthologues (right). Links between tree tips represent host–parasite associations. Conventions as in figure 1. Asterisk indicates taxa removed for comparative analysis. Caret indicates nodes collapsed in parasite tree for comparative analysis. World maps created using rnaturalearth (https://github.com/ropensci/rnaturalearth). (Online version in colour.)

material, table S2). However, based on divergence times obtained from the host trees (electronic supplementary material, figures S9–S14), these additional events would have occurred prior to our estimates for the onset of louse diversification and thus are incompatible with the time-calibrated host and parasite trees (i.e. the 95% confidence intervals did not overlap). Examining dove clades that are inferred to have colonized new biogeographic regions through long-distance dispersal, we observed two patterns: (i) doves acquiring lice from native hosts following dispersal and colonization of a new region (three cases) and (ii) a dove bringing lice to new region through dispersal, which then switch to a native host in that region (one case). Three examples of doves acquiring lice from native hosts after colonizing a new region include *Columba* species, which had a New World origin and acquired parasites after colonizing Africa; *Turtur* species, which had an Australasian origin and acquired lice after colonizing Africa; and *Macropygia* + *Reinwardtoena* species, which had a New World origin and acquired lice upon colonizing Australasia.

An example of doves bringing lice to new region includes lice carried from Africa to Australasia by dispersing *Columba* species, where the lice then switched to the genus *Chalcophaps* (electronic supplementary material, figures S15–S17).

## 4. Discussion

In this study, we explored new ecological opportunities provided by host dispersal for adaptive radiation in parasites. Specifically, we used phylogenomic datasets derived from whole-genome sequence reads for dove hosts and their wing louse parasites to reconstruct the biogeographic context and timing of host-switching events. We found that long-distance dispersal by doves was often followed by host-switching of lice to a newly sympatric host lineage. We also found that the contribution of host-switching to parasite speciation among extant lineages declined over time, relative to cospeciation. These results are compatible with the

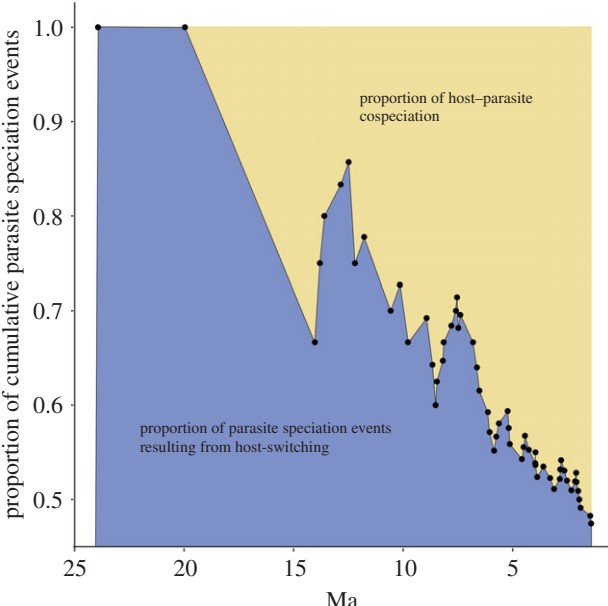

**Figure 3.** Proportion of cumulative host–parasite cospeciation events and parasite speciation events resulting from host-switching over time. Data based on cophylogenetic reconstructions of doves and wing lice that assumes an Australasian origin of parasitic wing lice with speciation events constrained by three time periods that correspond with host dispersal between geographical regions described in figure 1. Wing louse speciation event ages based on phylogenetic results described in the electronic supplementary material, figure S18. (Online version in colour.)

hypothesis that hosts without parasites may have provided new ecological opportunities for parasite speciation and as parasite-free host species declined in frequency, opportunities for host-switching declined as well. Alternatively, it may instead have been the case that ancestral dove lineages already harboured wing lice. In this case, host-switching would have led to multiple competing parasites on single-host species, with subsequent extinction of one parasite species, potentially due to competition. Experimental studies examining parasite establishment following a simulated host-switch, both in the presence or absence of potential competitors, would provide novel insights into whether parasite-free hosts make host-switching more likely than when a potential competitor is present.

Focusing specifically on the evolutionary history of the avian hosts, our phylogenomic results show that early dove diversification was shaped by continental drift and occurred prior to the diversification of extant wing louse species groups. Modern dove species diversity is highest in the southern hemisphere, with centres of dove diversity found in Australasia, Asia, South America and central Africa [40]. Our analyses point to a subset of these areas, specifically South America and Australasia, as the ancestral range of doves. Doves began to diversify just after the Cretaceous-Paleogene (K-Pg) boundary, when Australia and South America were connected via Antarctica. The Australia–Antarctica–South American land bridge had a near tropical climate during this period [41], which could have supported a diverse assemblage of early dove species across the region and allowed them to move among these continents [39,42]. Australia broke away from Antarctica at 49–50 Ma [43], followed by South America at 39 Ma [44]. The timing of this separation corresponds to initial divergence of dove lineages, with one lineage remaining in the New World and the other in Australasia.

Following the geologic breakup of the ancestral range of doves, further diversification was shaped by long-distance dispersal. The separation of Australia and South America from Antarctica marks the start of a period when doves could have used Antarctica as a stepping-stone in dispersal between Australia and South America. We found evidence of one such event, with cuckoo-doves (*Macropygia + Reinward-toena*) moving from the New World to Australasia at this time. The establishment of circumpolar ocean currents around Antarctica 33–34 Ma [45] substantially cooled the region [46] and would have prevented further use of this region as a stepping-stone between Australia and the New World [39]. During this later period of dispersal, we see evidence of the movement of green doves (*Treron*), emerald doves (*Chalcophaps*) and wood doves (*Turtur*) from Australasia to Eurasia and Africa. We also see evidence of movement from the New World to the Old World by a clade of rock (*Columba*) and collared (*Streptopelia*) doves (this lineage colonized Africa, Eurasia and Australasia).

While doves began to diversify shortly after the K-Pg boundary, wing louse speciation in the ancestors of extant lineages was delayed relative to host speciation and appears to have been facilitated by host-switching following host dispersal. It appears that extant wing louse species diversity was derived from a single lineage that was isolated within a single geographical region prior to diversification. Diversification among extant lineages of wing lice began at 15 or 24 Ma, with lice colonizing multiple continents. The timing of wing louse diversification corresponds with the later period of host dispersal. Thus, host dispersal appears to have provided ecological opportunity for host-switching by wing lice. In particular, our phylogenetic reconstructions indicate that resident wing lice switched to newly arriving dove lineages. We also see evidence of the reverse pattern (i.e. wing lice on newly arriving host lineages switching to resident doves).

Host-switching allowed wing lice to establish on new host clades, leading to parasite radiation. When examining extant taxa, our data show that parasite speciation events triggered by host-switching events are significantly older than host–parasite cospeciation events, highlighting the relative importance of dispersal followed by host-switching in wing louse diversification. The decline in the relative frequency of host-switching over time is consistent with congeneric competition [47] having diminished ecological opportunities for wing louse speciation by interfering with host-switching and establishment. However, experiments are needed to explore whether host-switching can be successful in the presence of a potentially competing species, in which case host-switching by extant lineages of lice may have drove other lineages extinct. A key finding from this work was, not that speciation diminished, but rather that cospeciation became a relatively more common mode of divergence. This shift was detectable by classifying the mechanism of parasite speciation (host-switching versus cospeciation) relative to time.

Our results suggest that long-distance dispersal and colonization of new biogeographic regions by host species is important for the diversification of parasites and that parasites may take advantage of new ecological opportunities on novel host species. We also infer that long-distance dispersal by host species can shape parasite diversity over evolutionary timescales, providing new opportunities for adaptive radiation by parasites. We observed parallels between our study system and another host–parasite

system, lice that parasitize humans. There is evidence that (i) modern humans acquired head lice from extinct hominids following dispersal and colonization of Eurasia [48], (ii) humans carried head lice with them as they colonized the New World via Beringia [49,50] and (iii) human lice switched to New World primates after humans colonized the New World [51]. These events mirror the evolutionary patterns of wing lice and their hosts, and suggest these processes may be broadly important for parasite diversification. Additional studies examining the consequences of long-distance host dispersal on parasite speciation and adaptive radiation are needed across a variety of parasites. The results of such studies may have broad relevance to the biology of invasive species and species conservation [18,19].

Ethics. Research on animals was conducted under University of Illinois, Champaign, Illinois IACUC protocols 10119, 13121, and 15212.

Data accessibility. Raw sequence data generated in this study are available from the Short Read Archive (https://www.ncbi.nlm.nih.gov/sra) and organized under BioProject PRJNA318048. Raw sequence data previously generated from parasites are also available from the Short Read Archive organized under BioProject PRJNA296666. Newly generated data have been made available on Figshare (https://doi.org/10.6084/m9.figshare.17108207) and include assembled orthologues, gene trees, ASTRAL tree, concatenation alignment, concatenation partition, RAxML-bipartition tree based on the concatenation alignment, base frequency by codon, MCMC trees, Jane input files, and sample collection data for both dove and louse species included in this study. The parasite tree used in this study was obtained from the Dryad Digital Repository: https://doi.org/10.5061/dryad.4812p [52] and is also available in the Figshare repository associated with this study [53].

Authors' contributions. B.M.B.: conceptualization, data curation, formal analysis, investigation, methodology, validation, visualization, writing—original draft and writing—review and editing; N.-P.N.: formal analysis, methodology, validation and writing—review and editing; J.M.A.: formal analysis, methodology, validation and writing—review and editing; R.M.W.: formal analysis and writing—review and editing; K.B.V.: formal analysis and writing—review and editing; A.D.S.: formal analysis, methodology, visualization and writing—review and editing; D.H.C.: conceptualization, methodology, resources, visualization, writing—original draft and writing—review and editing; S.E.B.: conceptualization, methodology, resources, visualization, writing—original draft and writing—review and editing; M.D.S.: conceptualization, methodology, resources and writing—review and editing; K.P.J.: conceptualization, data curation, formal analysis, funding acquisition, investigation, methodology, project administration, resources, supervision, validation, visualization, writing—original draft and writing—review and editing.

All authors gave final approval for publication and agreed to be held accountable for the work performed therein.

Competing interests. We declare we have no competing interests.

Funding. This work was supported by the US National Science Foundation grant nos DEB-1342604, DEB-1926919 and DEB-1925487 to K.P.J.; DEB-1342600 to D.H.C., S.E.B. and M.D.S.; DEB-9703003 and DEB-0107947 to D.H.C.; DEB-1149160 to M.D.S.; and DEB-1925312 to J.M.A.; US National Institutes of Health grant nos R01GM115996 and R35GM131787 to M.D.S.; Swiss National Science Foundation grant nos PP00P3_170664 and PP00P3_202669 to R.M.W.; and the Virginia Commonwealth University, Department of Life Sciences.

Acknowledgements. We thank the following institutions for providing samples of dove tissues used in this study: Field Museum of Natural History, University of Kansas Museum of Natural History, Louisiana State University Museum of Natural Science, University of Washington Burke Museum, Australian National Wildlife Collection and the US National Museum of Natural History. We thank David Steadman of the Florida Museum of Natural History for help interpreting fossil evidence, and James Baldwin-Brown of the University of Utah for helpful advice. We thank the anonymous reviewers for their constructive comments on the original manuscript. High Performance Computing resources were provided by the High Performance Research Computing (HPRC) Core Facility at Virginia Commonwealth University (http://chipc.vcu.edu) were used for conducting the research reported in this work.

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
