## [Peer Review File · Proceedings of the Royal Society B: Biological Sciences]

Review History

RSPB-2021-1759.R0 (Original submission)

Review form: Reviewer 1

Recommendation

Accept with minor revision (please list in comments)

Scientific importance: Is the manuscript an original and important contribution to its field?

Good

General interest: Is the paper of sufficient general interest?

Excellent

Quality of the paper: Is the overall quality of the paper suitable?

Good

Is the length of the paper justified?

Yes

Should the paper be seen by a specialist statistical reviewer?

No

Do you have any concerns about statistical analyses in this paper? If so, please specify them explicitly in your report.

No

It is a condition of publication that authors make their supporting data, code and materials available - either as supplementary material or hosted in an external repository. Please rate, if applicable, the supporting data on the following criteria.

Is it accessible?

Yes

Is it clear?

Yes

Is it adequate?

Yes

Do you have any ethical concerns with this paper?

No

Comments to the Author

The paper describes the phylogeographic history of pigeons and doves and maps it to that of their wing lice. The authors try to differentiate between events of host-switching and adaptive radiation and decipher the course of events through timed phylogenies and cophylogeny and phylogeography analyses, based on large datasets. The paper is well written and interesting to a broad range of readers from parasitology, ecology, evolutionary biology and general biology. It is of relevance to the biology of invasive species and species conservation.

I have only a few comments and mainly request more details in methods and higher accuracy in documentation (e.g. figure legends). Please find my points below (major and minor mixed):

- The details on the time calibration of the lice tree are missing from material and methods. I know that fossils are a problem in the Phthiraptera but in contrast to the host tree, it is unclear how the parasite tree was dated. Some timing information is in Figures S5-S8 but there is no references to the origin of this data. How were the dates specified in these supplementary trees calculated? The most basal lineage in the lice tree has a species from Australia and one from South America, with relatively long branches. It may be a coincidence but this nicely fits the basal split in the host tree.

- Cophylogeny: It seems a bit risky to base all conclusions on the analyses of only one cophylogeny method. Additionally to the event-based method (Jane) I think you should apply a global fit approach (APE or best the advanced PACO R package). In event-based methods, the estimation of congruence between the observed trees is typically based on maximizing cospeciation events, and more so if you have polytomies (by the way to collapse nodes with less than 75% bootstrap support seems a bit excessive, >85 or >90 would be more feasible in a multi gene phylogenetic tree). Such estimates are not always biologically realistic. Global fit methods allow the evaluation of phylogenetic uncertainty and the contribution of individual host-symbiont links to overall congruence. This may also be important in the light of interpretations regarding an ancestral host (see below).

- In Figures S9-S11 the legend does not include an explanation of the node colors and dots vs circles. What is the likelihood for each of these scenarios to occur (Jane)?

- Wing lice diversified 15 or 24 mya while doves diversified much earlier (51 or 60 mya). It is hence interesting to estimate where and when the parasites came into play and who the ancestral host/parasite pair could have been. Cophylogeny analysis supports four (!) possible origins on three different continents, however, the basal lineage in the parasite tree indicates that

this is likely either Australia or South America. Maybe this could be listed as supportive information. It is unfortunate that other more basal nodes have such low support. Weighed contributions of cophylogenetic signal obtained from PACO could also help to better elucidate the parasites' origin.

- Table S1: the numbering system of divergent lineages is according to Boyd et al. [16]. Reference 16 is inexistent in the supplementary data file and is Alavaex et al. 2010 in the main manuscript. I guess you refer to Boyd et al. 2017 [24 in main manuscript]? However, the lineages in this paper are labelled with letters not numbers as in table S1. Please clarify. On another note, please specify methods that led to the results in table headers of tables S1 and S2.
- The left section of fig. 2 is a repetition of fig. 1. While this may be necessary to illustrate cophylogeny relationships, I am not a fan of tangled trees and believe that the interpretations on what (host dispersal, host switching, parasite radiation) happened in which order on which continent could nicely be illustrated in an additional figure (map with arrows for introductions and symbols for radiation etc.) and would help the reader to understand the historic course of interaction between different host and parasite species.
- Spell out K-Pg boundary at first mention

Review form: Reviewer 2

Recommendation

Major revision is needed (please make suggestions in comments)

Scientific importance: Is the manuscript an original and important contribution to its field?

Excellent

General interest: Is the paper of sufficient general interest?

Excellent

Quality of the paper: Is the overall quality of the paper suitable?

Good

Is the length of the paper justified?

Yes

Should the paper be seen by a specialist statistical reviewer?

No

Do you have any concerns about statistical analyses in this paper? If so, please specify them explicitly in your report.

No

It is a condition of publication that authors make their supporting data, code and materials available - either as supplementary material or hosted in an external repository. Please rate, if applicable, the supporting data on the following criteria.

Is it accessible?

Yes

Is it clear?

Yes

Is it adequate?

Yes

Do you have any ethical concerns with this paper?

No

Comments to the Author

Review of RSPB-2021-1759

This is a really nice manuscript based on timely questions, robust analyses and generally sound interpretations. There are, however, some general issues that, in my opinion, require discussion / clarification, particularly for the non-expert in dove-feather lice associations. Please note the following:

1. The authors consider just two co-evolutionary mechanisms, i.e., cospeciation and host-switching, as drivers of the observed patterns of association of the system under study. I wonder why other processes, e.g., missing-the-boat, duplications, or simple co-accommodation (failure to speciate) were not considered in co-phylogenetic reconstructions. This should be clarified.

2. I am not particularly familiar with the degree of specificity of feather lice, but in other systems (e.g., marine helminths and their hosts) there is plenty of new host colonization events without concurrent speciation of the parasite (i.e., co-accommodation). I am aware that co-adaptation of feather lice to new hosts must, in principle, be more stringent, and this might generate speciation as a side-effect. However, this does not necessarily mean that all new host colonizations result in lice speciation, or does it? This should be clarified. More generally, I would like to see a bit of clarification on the assumption (made in this and many other studies of similar nature) that host-parasite associations are univocal (i.e., one parasite species – one host species). If there is even a moderate number of instances in which the same feather louse species occur on more than a dove species, I'd like to know how this phenomenon fits in the relative importance of host-switching vs. cospeciation during radiations.

3. The authors suggest that the saturation of niche space for host-switching is the explanation for the progressive preponderance of cospeciation over time as a mechanism generating diversity. However, this assumption is not really tested and appears to be taken for granted, which is debatable. First of all, the study deals with just 61 (which is, in any event, to be praised!) of the c. 357 spp. of Columbiformes, i.e., we have a partial picture of the history of associations and therefore little idea of how the unknown portion of the history might have influenced patterns of the studied part. More importantly, is there any robust evidence of niche saturation in feather lice? History is by definition idiosyncratic, and a simple phenomenon of pool exhaustion could have resulted in a reduction of host-switching events over time. Besides, the relative importance of cospeciation also depends of the rate of host speciation itself (not to mention other evolutionary phenomena not considered in the analyses –see my first comment). Therefore, it is difficult to make generalizations (or predictions) beforehand. In short, I would recommend the authors to 'tone down' their claims on the relative importance of niche saturation and adopt a more pluralistic set of potential explanations (or at least support better their selected explanation).

Congratulations on a really nice work.

Decision letter (RSPB-2021-1759.R0)

23-Sep-2021

Dear Dr Boyd:

I am writing to inform you that your manuscript RSPB-2021-1759 entitled "Long-distance dispersal of pigeons and doves generated new ecological opportunities for host-switching and adaptive radiation by their parasites." has, in its current form, been rejected for publication in Proceedings B.

This action has been taken on the advice of referees, who have recommended that substantial revisions are necessary. With this in mind we would be happy to consider a resubmission, provided the comments of the referees are fully addressed. However please note that this is not a provisional acceptance.

Sincerely,
Professor Hans Heesterbeek
mailto: proceedingsb@royalsociety.org

Associate Editor
Board Member: 1
Comments to Author:

Your manuscript has received two expert reviews, which raise both general points and specific analytical points about your study. I have gone through these reviews and your manuscript, and believe that all of the issues raised by the reviewers need to be addressed, including the additional analyses. In any revision, please make sure to include a point-by-point response to the reviewer comments, and bear in mind that if you need to explain something in response to the reviewer comments it will also need to be explained in the revised manuscript so that future readers don't have the same questions.

Reviewer(s)' Comments to Author:
Referee: 1

Comments to the Author(s)

The paper describes the phylogeographic history of pigeons and doves and maps it to that of their wing lice. The authors try to differentiate between events of host-switching and adaptive radiation and decipher the course of events through timed phylogenies and cophylogeny and phylogeography analyses, based on large datasets. The paper is well written and interesting to a

broad range of readers from parasitology, ecology, evolutionary biology and general biology. It is of relevance to the biology of invasive species and species conservation.

I have only a few comments and mainly request more details in methods and higher accuracy in documentation (e.g. figure legends). Please find my points below (major and minor mixed):

- The details on the time calibration of the lice tree are missing from material and methods. I know that fossils are a problem in the Phthiraptera but in contrast to the host tree, it is unclear how the parasite tree was dated. Some timing information is in Figures S5-S8 but there is no references to the origin of this data. How were the dates specified in these supplementary trees calculated? The most basal lineage in the lice tree has a species from Australia and one from South America, with relatively long branches. It may be a coincidence but this nicely fits the basal split in the host tree.

- Cophylogeny: It seems a bit risky to base all conclusions on the analyses of only one cophylogeny method. Additionally to the event-based method (Jane) I think you should apply a global fit approach (APE or best the advanced PACO R package). In event-based methods, the estimation of congruence between the observed trees is typically based on maximizing cospeciation events, and more so if you have polytomies (by the way to collapse nodes with less than 75% bootstrap support seems a bit excessive, >85 or >90 would be more feasible in a multi gene phylogenetic tree). Such estimates are not always biologically realistic. Global fit methods allow the evaluation of phylogenetic uncertainty and the contribution of individual host-symbiont links to overall congruence. This may also be important in the light of interpretations regarding an ancestral host (see below).

- In Figures S9-S11 the legend does not include an explanation of the node colors and dots vs circles. What is the likelihood for each of these scenarios to occur (Jane)?

- Wing lice diversified 15 or 24 mya while doves diversified much earlier (51 or 60 mya). It is hence interesting to estimate where and when the parasites came into play and who the ancestral host/parasite pair could have been. Cophylogeny analysis supports four (!) possible origins on three different continents, however, the basal lineage in the parasite tree indicates that this is likely either Australia or South America. Maybe this could be listed as supportive information. It is unfortunate that other more basal nodes have such low support. Weighed contributions of cophylogenetic signal obtained from PACO could also help to better elucidate the parasites' origin.

- Table S1: the numbering system of divergent lineages is according to Boyd et al. [16]. Reference 16 is inexistent in the supplementary data file and is Alavaex et al. 2010 in the main manuscript. I guess you refer to Boyd et al. 2017 [24 in main manuscript]? However, the lineages in this paper are labelled with letters not numbers as in table S1. Please clarify. On another note, please specify methods that led to the results in table headers of tables S1 and S2.

- The left section of fig. 2 is a repetition of fig. 1. While this may be necessary to illustrate cophylogeny relationships, I am not a fan of tangled trees and believe that the interpretations on what (host dispersal, host switching, parasite radiation) happened in which order on which continent could nicely be illustrated in an additional figure (map with arrows for introductions and symbols for radiation etc.) and would help the reader to understand the historic course of interaction between different host and parasite species.

- Spell out K-Pg boundary at first mention

Referee: 2

Comments to the Author(s)

Review of RSPB-2021-1759

This is a really nice manuscript based on timely questions, robust analyses and generally sound interpretations. There are, however, some general issues that, in my opinion, require discussion / clarification, particularly for the non-expert in dove-feather lice associations. Please note the following:

1. The authors consider just two co-evolutionary mechanisms, i.e., cospeciation and host-switching, as drivers of the observed patterns of association of the system under study. I wonder

why other processes, e.g., missing-the-boat, duplications, or simple co-accommodation (failure to speciate) were not considered in co-phylogenetic reconstructions. This should be clarified.

2. I am not particularly familiar with the degree of specificity of feather lice, but in other systems (e.g., marine helminths and their hosts) there is plenty of new host colonization events without concurrent speciation of the parasite (i.e., co-accommodation). I am aware that co-adaptation of feather lice to new hosts must, in principle, be more stringent, and this might generate speciation as a side-effect. However, this does not necessarily mean that all new host colonizations result in lice speciation, or does it? This should be clarified. More generally, I would like to see a bit of clarification on the assumption (made in this and many other studies of similar nature) that host-parasite associations are univocal (i.e., one parasite species – one host species). If there is even a moderate number of instances in which the same feather louse species occur on more than a dove species, I'd like to know how this phenomenon fits in the relative importance of host-switching vs. cospeciation during radiations.

3. The authors suggest that the saturation of niche space for host-switching is the explanation for the progressive preponderance of cospeciation over time as a mechanism generating diversity. However, this assumption is not really tested and appears to be taken for granted, which is debatable. First of all, the study deals with just 61 (which is, in any event, to be praised!) of the c. 357 spp. of Columbiformes, i.e., we have a partial picture of the history of associations and therefore little idea of how the unknown portion of the history might have influenced patterns of the studied part. More importantly, is there any robust evidence of niche saturation in feather lice? History is by definition idiosyncratic, and a simple phenomenon of pool exhaustion could have resulted in a reduction of host-switching events over time. Besides, the relative importance of cospeciation also depends of the rate of host speciation itself (not to mention other evolutionary phenomena not considered in the analyses –see my first comment). Therefore, it is difficult to make generalizations (or predictions) beforehand. In short, I would recommend the authors to 'tone down' their claims on the relative importance of niche saturation and adopt a more pluralistic set of potential explanations (or at least support better their selected explanation).

Congratulations on a really nice work.

Author's Response to Decision Letter for (RSPB-2021-1759.R0)

See Appendix A.

RSPB-2021-2306.R0

Review form: Reviewer 2

Recommendation

Accept as is

Scientific importance: Is the manuscript an original and important contribution to its field?

Excellent

General interest: Is the paper of sufficient general interest?

Excellent

Quality of the paper: Is the overall quality of the paper suitable?

Excellent

Is the length of the paper justified?

Yes

Should the paper be seen by a specialist statistical reviewer?

No

Do you have any concerns about statistical analyses in this paper? If so, please specify them explicitly in your report.

No

It is a condition of publication that authors make their supporting data, code and materials available - either as supplementary material or hosted in an external repository. Please rate, if applicable, the supporting data on the following criteria.

Is it accessible?

Yes

Is it clear?

Yes

Is it adequate?

Yes

Do you have any ethical concerns with this paper?

No

Comments to the Author

The authors have satisfactorily addressed all my queries. I think the paper should now be published.

Review form: Reviewer 3

Recommendation

Reject - article is scientifically unsound

Scientific importance: Is the manuscript an original and important contribution to its field?

Good

General interest: Is the paper of sufficient general interest?

Good

Quality of the paper: Is the overall quality of the paper suitable?

Marginal

Is the length of the paper justified?

Yes

Should the paper be seen by a specialist statistical reviewer?

No

Do you have any concerns about statistical analyses in this paper? If so, please specify them explicitly in your report.

Yes

It is a condition of publication that authors make their supporting data, code and materials available - either as supplementary material or hosted in an external repository. Please rate, if applicable, the supporting data on the following criteria.

Is it accessible?

Yes

Is it clear?

Yes

Is it adequate?

No

Do you have any ethical concerns with this paper?

Yes

Comments to the Author

This is a great phylogenomic time-tree of the Columbidae, with a useful biogeographic analysis that shows strong signal of global biogeographic regions. The corresponding wing-louse phylogeny indicates lots of co-speciation. Those results are exciting. A central claim of the paper relates to increasing number of co-speciation events, relative to host-switch events, toward the present. This result is not convincing because it is not appropriately described or tested with respect to a null expectation (see additional details that I describe below). The paper also fails on issues of methods and sample documentation, and the lack of documentation should be easily remedied, but the issue of uncertainty in the louse time tree may be a serious one affecting the interpretation of results.

The assertion that older dispersal events by doves led to host switching, followed by predominance of younger cospeciation events is central to the paper, the main result reported in the abstract. However, it's problematic in multiple ways. Dove 'dispersal' as the specific driver of ancient inferred host-switch events is a bold assertion because of the great deal of uncertainty about ancient ranges, known range dynamism (i.e. expansion and contraction as alternative to 'dispersal') of many bird groups through the course of the Cenozoic, and other potential sources of phylogenetic conflict that could cause the appearance of host switching – these caveats should be noted, and assertions made more conservative -- a better fit to the evidence at hand would be to assert 'apparent host-switch events' without attributing them to dispersal specifically.

Methods section: where is the description of the louse molecular data and tree? Except the note about 977 loci in the caption of Fig. 2, I couldn't find it (another tiny note suggesting it came from Dryad?). The time-dependence of many of the results means that more attention needs to be given to the louse fossil calibrations and the uncertainty associated with them. The description given indicates that there are mostly secondary calibrations involved – the basis for these needs to be described. What assumptions were made, based on what prior data and analyses, and where is the degree of uncertainty described? (this reviewer is satisfied with the avian timetree because of the substantial previous work on that, and general concordance with fossil record). Uncertainty of node timing is not depicted anywhere – but it should be included with timetrees by default – to not do so is unhelpful at best. Table S1 is inadequate because it neither contains useful information about the basis for the parasite calibrations, nor information about how the calibration point was used (max? min? what prior distributions?).

Line 260: "The median age of louse speciation events due to cospeciation was significantly younger than those due to host-switching (Wilcoxon Signed Rank test $P = 0.021$)." This weak

statistical result is carrying a lot of weight in this paper (central result described in abstract), yet I assert that it is invalid for a simple reason: each host-switch event can erase the signal of cospeciation events that preceded it; therefore, ancient cospeciation events are more likely to have been erased by subsequent evolution; therefore, a null model under constant rates of cospeciation and host switching over time is likely to lead to the appearance of increased rates of cospeciation toward the present.

Lines 263-265: The results are said to be “incompatible” with the time trees, but was error properly carried and reported for the louse time tree? How is ‘incompatibility’ quantified, and what is the degree of ‘incompatibility’?

Line 286: “as parasite-free host species declined in frequency” That’s a bold inference to make – can’t host switching happen even by displacing a current louse occupant? What proportion of extant dove species are ‘parasite-free’? If there’s no evidence for this specific claim, the following claim also is an overreach: “These results indicate that hosts without parasites may have provided new ecological opportunities for parasite adaptive radiation.”. Alternatively, ancient host-switches reflect the outcome of competition...

Line 333-4: “It appears that a single wing louse lineage was isolated within a single geographical region prior to diversification.” Appearances are deceiving... Just as coalescence of a gene tree to a single individual in the past does not imply a population size of one... other lice may have been on other doves, they may have simply failed to leave extant descendants! In other words, I don’t think it’s OK to make an (arguably) implausible ecological hypothesis simply because of an absence of evidence. It appears that all or nearly all of these 61 columbid species are ‘infested’, therefore it’s parsimonious to suggest that all columbid species have always been ‘infested’, with periodic extinctions driven by host-switching.

Line 348-9: “However, other processes could conceivably have resulted in this pattern. For example, an increase in the rate of host diversification could have provided additional opportunities for cospeciation, or the opportunity for host-switching could have diminished over time.” This implies that the method by which events were quantified did not correct for ‘opportunity’. If this is the case we should always expect increasing co-speciation toward the present, by default. The authors should perhaps return to the white board to reconsider the appropriate way to identify this supposed pattern.

Line 350-1: “(e.g. by a decline in the abundance of hippoboscids, which can vector wing lice between host species [48])” There is no cited evidence here for a such a decline in hippoboscids (I looked up and read reference 48 to check), making this entirely speculative – please delete it.

The sample table is buried deeply in the Dryad accession among many small files that document minor components of the analyses. It is also deeply inadequate, with hodge-podge sample identifiers and formats (use Darwin Core format where possible, and/or at least identify the identifier type, whether personal catalog number or tissue collection number, bird collection number, etc.), many blank cells, and institutions of origin not clearly indicated. Catalog numbers from IL natural history survey attributed to samples loaned from other collections are also present – I doubt that curators of those collections would be OK with these additional ascribed identifiers, but the authors here at least have the obligation to provide clarification. I further strongly suggest that this table be included (or repeated) in the supplement of the present paper, if not the main paper itself. Finally, where is the sample table for the lice?... same principles of documentation apply.

The discussion of the lice as an ‘adaptive radiation’ is a confusing and unnecessary to this paper.

Biogeobears analysis is poorly documented – many very impactful parameter and model settings are not considered or described.

Decision letter (RSPB-2021-2306.R0)

18-Nov-2021

I am writing to inform you that this version of your manuscript RSPB-2021-2306 entitled "Long-distance dispersal of pigeons and doves generated new ecological opportunities for host-switching and adaptive radiation by their parasites." has, in its current form, been rejected for publication in Proceedings B.

This action has been taken on the advice of a new referee, who has indicated that revisions are necessary. With this in mind we would be happy to consider a resubmission, provided the comments of the referee are fully addressed. However please note that this is not a provisional acceptance. As explained by the Associate Editor in their comments below, only one of the original reviewers was available to assess your revision. We have therefore sought an additional reviewer. This reviewer raises several issues and because the reviewer is new, it is fair that you are given the opportunity to respond to the criticism in a further revision.

The resubmission will be treated as a new manuscript. However, we will approach the same reviewers if they are available and it is deemed appropriate to do so by the Associate Editor. Please note that resubmissions must be submitted within six months of the date of this email. In exceptional circumstances, extensions may be possible if agreed with the Editorial Office. Manuscripts submitted after this date will be automatically rejected.

Please find below the comments made by the referees, not including confidential reports to the Editor, which I hope you will find useful.

- 1) A 'response to referees' document including details of how you have responded to the comments, and the adjustments you have made.
- 2) A clean copy of the manuscript and one with 'tracked changes' indicating your 'response to referees' comments document.
- 3) Line numbers in your main document.
- 4) Please read our data sharing policies to ensure that you meet our requirements <https://royalsociety.org/journals/authors/author-guidelines/#data>.

Sincerely,
Professor Hans Heesterbeek
<mailto:proceedingsb@royalsociety.org>

Associate Editor Board Member
Comments to Author:

Thank you for revising your manuscript in response to comments from the previous reviewers. Because only one of these was able to review your modified manuscript, we had to send it out to a new reviewer, and they have identified a series of potential significant issues with your analyses and interpretations. Any revision of your manuscript needs to engage with these comments, many of which require either additional analyses or a higher level of clarity in the

documentation of your work. I hope that you are able to address satisfactorily all of their criticisms.

Reviewer(s)' Comments to Author:

Referee: 2

Comments to the Author(s).

The authors have satisfactorily addressed all my queries. I think the paper should now be published.

Referee: 3

Comments to the Author(s).

This is a great phylogenomic time-tree of the Columbidae, with a useful biogeographic analysis that shows strong signal of global biogeographic regions. The corresponding wing-louse phylogeny indicates lots of co-speciation. Those results are exciting. A central claim of the paper relates to increasing number of co-speciation events, relative to host-switch events, toward the present. This result is not convincing because it is not appropriately described or tested with respect to a null expectation (see additional details that I describe below). The paper also fails on issues of methods and sample documentation, and the lack of documentation should be easily remedied, but the issue of uncertainty in the louse time tree may be a serious one affecting the interpretation of results.

The assertion that older dispersal events by doves led to host switching, followed by predominance of younger cospeciation events is central to the paper, the main result reported in the abstract. However, it's problematic in multiple ways. Dove 'dispersal' as the specific driver of ancient inferred host-switch events is a bold assertion because of the great deal of uncertainty about ancient ranges, known range dynamism (i.e. expansion and contraction as alternative to 'dispersal') of many bird groups through the course of the Cenozoic, and other potential sources of phylogenetic conflict that could cause the appearance of host switching – these caveats should be noted, and assertions made more conservative -- a better fit to the evidence at hand would be to assert 'apparent host-switch events' without attributing them to dispersal specifically.

Methods section: where is the description of the louse molecular data and tree? Except the note about 977 loci in the caption of Fig. 2, I couldn't find it (another tiny note suggesting it came from Dryad?). The time-dependence of many of the results means that more attention needs to be given to the louse fossil calibrations and the uncertainty associated with them. The description given indicates that there are mostly secondary calibrations involved – the basis for these needs to be described. What assumptions were made, based on what prior data and analyses, and where is the degree of uncertainty described? (this reviewer is satisfied with the avian timetree because of the substantial previous work on that, and general concordance with fossil record). Uncertainty of node timing is not depicted anywhere – but it should be included with timetrees by default – to not do so is unhelpful at best. Table S1 is inadequate because it neither contains useful information about the basis for the parasite calibrations, nor information about how the calibration point was used (max? min? what prior distributions?).

Line 260: "The median age of louse speciation events due to cospeciation was significantly younger than those due to host-switching (Wilcoxon Signed Rank test $P = 0.021$)." This weak statistical result is carrying a lot of weight in this paper (central result described in abstract), yet I assert that it is invalid for a simple reason: each host-switch event can erase the signal of cospeciation events that preceded it; therefore, ancient cospeciation events are more likely to have been erased by subsequent evolution; therefore, a null model under constant rates of cospeciation and host switching over time is likely to lead to the appearance of increased rates of cospeciation toward the present.

Lines 263-265: The results are said to be “incompatible” with the time trees, but was error properly carried and reported for the louse time tree? How is ‘incompatibility’ quantified, and what is the degree of ‘incompatibility’?

Line 286: “as parasite-free host species declined in frequency” That’s a bold inference to make – can’t host switching happen even by displacing a current louse occupant? What proportion of extant dove species are ‘parasite-free’? If there’s no evidence for this specific claim, the following claim also is an overreach: “These results indicate that hosts without parasites may have provided new ecological opportunities for parasite adaptive radiation.”. Alternatively, ancient host-switches reflect the outcome of competition...

Line 333-4: “It appears that a single wing louse lineage was isolated within a single geographical region prior to diversification.” Appearances are deceiving... Just as coalescence of a gene tree to a single individual in the past does not imply a population size of one... other lice may have been on other doves, they may have simply failed to leave extant descendants! In other words, I don’t think it’s OK to make an (arguably) implausible ecological hypothesis simply because of an absence of evidence. It appears that all or nearly all of these 61 columbid species are ‘infested’, therefore it’s parsimonious to suggest that all columbid species have always been ‘infested’, with periodic extinctions driven by host-switching.

Line 348-9: “However, other processes could conceivably have resulted in this pattern. For example, an increase in the rate of host diversification could have provided additional opportunities for cospeciation, or the opportunity for host-switching could have diminished over time.” This implies that the method by which events were quantified did not correct for ‘opportunity’. If this is the case we should always expect increasing co-speciation toward the present, by default. The authors should perhaps return to the white board to reconsider the appropriate way to identify this supposed pattern.

Line 350-1: “(e.g. by a decline in the abundance of hippoboscids, which can vector wing lice between host species [48])” There is no cited evidence here for a such a decline in hippoboscids (I looked up and read reference 48 to check), making this entirely speculative – please delete it.

The sample table is buried deeply in the Dryad accession among many small files that document minor components of the analyses. It is also deeply inadequate, with hodge-podge sample identifiers and formats (use Darwin Core format where possible, and/or at least identify the identifier type, whether personal catalog number or tissue collection number, bird collection number, etc.), many blank cells, and institutions of origin not clearly indicated. Catalog numbers from IL natural history survey attributed to samples loaned from other collections are also present – I doubt that curators of those collections would be OK with these additional ascribed identifiers, but the authors here at least have the obligation to provide clarification. I further strongly suggest that this table be included (or repeated) in the supplement of the present paper, if not the main paper itself. Finally, where is the sample table for the lice?... same principles of documentation apply.

The discussion of the lice as an ‘adaptive radiation’ is a confusing and unnecessary to this paper.

Biogeobears analysis is poorly documented – many very impactful parameter and model settings are not considered or described.

Author's Response to Decision Letter for (RSPB-2021-2306.R0)

See Appendix B.

RSPB-2022-0042.R0

Review form: Reviewer 3

Recommendation

Accept as is

Scientific importance: Is the manuscript an original and important contribution to its field?

Good

General interest: Is the paper of sufficient general interest?

Good

Quality of the paper: Is the overall quality of the paper suitable?

Good

Is the length of the paper justified?

Yes

Should the paper be seen by a specialist statistical reviewer?

No

Do you have any concerns about statistical analyses in this paper? If so, please specify them explicitly in your report.

No

It is a condition of publication that authors make their supporting data, code and materials available - either as supplementary material or hosted in an external repository. Please rate, if applicable, the supporting data on the following criteria.

Is it accessible?

Yes

Is it clear?

Yes

Is it adequate?

Yes

Do you have any ethical concerns with this paper?

No

Comments to the Author

I'm impressed with the response to reviewers and changes made in revision. I'm grateful for the authors thorough responses, particularly with respect to the issue about relative timing of host-switch and co-speciation events. I generally agree with the main arguments they put forward and I'm also gratified that they improved their explanations in the paper by adding appropriate caveats and qualifications. Regarding sample documentation -- it's greatly improved, and meets the standards of the field. My congratulations to the authors on completion of this ambitious study.

Decision letter (RSPB-2022-0042.R0)

10-Feb-2022

Dear Dr Boyd

I am pleased to inform you that your Review manuscript RSPB-2022-0042 entitled "Long-distance dispersal of pigeons and doves generated new ecological opportunities for host-switching and adaptive radiation by their parasites." has been accepted for publication in Proceedings B.

The referee does not recommend any further changes. Therefore, please proof-read your manuscript carefully and upload your final files for publication. Because the schedule for publication is very tight, it is a condition of publication that you submit the revised version of your manuscript within 7 days. If you do not think you will be able to meet this date please let me know immediately.

To upload your manuscript, log into <http://mc.manuscriptcentral.com/prsb> and enter your Author Centre, where you will find your manuscript title listed under "Manuscripts with Decisions." Under "Actions," click on "Create a Revision." Your manuscript number has been appended to denote a revision.

You will be unable to make your revisions on the originally submitted version of the manuscript. Instead, upload a new version through your Author Centre.

- 1) A text file of the manuscript (doc, txt, rtf or tex), including the references, tables (including captions) and figure captions. Please remove any tracked changes from the text before submission. PDF files are not an accepted format for the "Main Document".
- 2) A separate electronic file of each figure (tiff, EPS or print-quality PDF preferred). The format should be produced directly from original creation package, or original software format. Please note that PowerPoint files are not accepted.
- 3) Electronic supplementary material: this should be contained in a separate file from the main text and the file name should contain the author's name and journal name, e.g. `authorname_procb_ESM_figures.pdf`

All supplementary materials accompanying an accepted article will be treated as in their final form. They will be published alongside the paper on the journal website and posted on the online figshare repository. Files on figshare will be made available approximately one week before the accompanying article so that the supplementary material can be attributed a unique DOI. Please see: <https://royalsociety.org/journals/authors/author-guidelines/>

4) Data-Sharing and data citation

It is a condition of publication that data supporting your paper are made available. Data should be made available either in the electronic supplementary material or through an appropriate repository. Details of how to access data should be included in your paper. Please see <https://royalsociety.org/journals/ethics-policies/data-sharing-mining/> for more details.

<http://datadryad.org/submit?journalID=RSPB&manu=RSPB-2022-0042> which will take you to your unique entry in the Dryad repository.

Once again, thank you for submitting your manuscript to Proceedings B and I look forward to receiving your final version. If you have any questions at all, please do not hesitate to get in touch.

Sincerely,
Professor Hans Heesterbeek
<mailto:proceedingsb@royalsociety.org>

Associate Editor
Board Member
Comments to Author:

Thanks for your engagement with the review process, which I believe has resulted in a much stronger, clearer, and more nuanced manuscript. It makes a significant contribution to the field, and I am delighted to recommend its acceptance for publication.

Reviewer(s)' Comments to Author:

Referee: 3

Comments to the Author(s).

I'm impressed with the response to reviewers and changes made in revision. I'm grateful for the authors thorough responses, particularly with respect to the issue about relative timing of host-switch and co-speciation events. I generally agree with the main arguments they put forward and I'm also gratified that they improved their explanations in the paper by adding appropriate caveats and qualifications. Regarding sample documentation -- it's greatly improved, and meets the standards of the field. My congratulations to the authors on completion of this ambitious study.

Sincerely,
Proceedings B
<mailto:proceedingsb@royalsociety.org>

Decision letter (RSPB-2022-0042.R1)

14-Feb-2022

Dear Dr Boyd

I am pleased to inform you that your manuscript entitled "Long-distance dispersal of pigeons and doves generated new ecological opportunities for host-switching and adaptive radiation by their parasites." has been accepted for publication in Proceedings B.

Your article has been estimated as being 9 pages long. Our Production Office will be able to confirm the exact length at proof stage.

Data Accessibility section

Open Access

Paper charges

Sincerely,

Appendix A

We thank the reviewers for their thoughtful review of our manuscript and for their helpful suggestions. We have made significant improvements to the manuscript based on their recommendations and provided a response to each reviewer comment.

Reviewer(s)' Comments to Author:

Referee: 1

Comments to the Author(s)

The paper describes the phylogeographic history of pigeons and doves and maps it to that of their wing lice. The authors try to differentiate between events of host-switching and adaptive radiation and decipher the course of events through timed phylogenies and cophylogeny and phylogeography analyses, based on large datasets. The paper is well written and interesting to a broad range of readers from parasitology, ecology, evolutionary biology and general biology. It is of relevance to the biology of invasive species and species conservation. I have only a few comments and mainly request more details in methods and higher accuracy in documentation (e.g. figure legends). Please find my points below (major and minor mixed):

RESPONSE: *Reviewer 1 suggested an additional analysis, which we have completed and included the results in the manuscript. This reviewer also suggested clarifying figure legends and changes to figure 2, which we address.*

- The details on the time calibration of the lice tree are missing from material and methods. I know that fossils are a problem in the Phthiraptera but in contrast to the host tree, it is unclear how the parasite tree was dated. Some timing information is in Figures S5-S8 but there is no references to the origin of this data. How were the dates specified in these supplementary trees calculated? The most basal lineage in the lice tree has a species from Australia and one from South America, with relatively long branches. It may be a coincidence but this nicely fits the basal split in the host tree.

RESPONSE: *Our apologies that the sources of these calibration points were not clear. The calibration points and their sources were listed in table S2. The basal calibration points of 108MYA and 66MYA were obtained from Smith et al. 2011 (Biol Lett 7:782) and Johnson et al. 2018 (Biol Lett 14:20180141). We have added informative text and references to figure legends for figures S5-S8. In the second part of this comment regarding the basal split, we suspect the reviewer is referring to the split of *C. triangularis* and *C. fortis*. This pattern is actually present throughout the tree, but the species divergence time estimates suggest this speciation occurred well after the Australasian-New World split in doves.*

- Cophylogeny: It seems a bit risky to base all conclusions on the analyses of only one cophylogeny method. Additionally to the event-based method (Jane) I think you

should apply a global fit approach (APE or best the advanced PACO R package). In event-based methods, the estimation of congruence between the observed trees is typically based on maximizing cospeciation events, and more so if you have polytomies (by the way to collapse nodes with less than 75% bootstrap support seems a bit excessive, >85 or >90 would be more feasible in a multi gene phylogenetic tree). Such estimates are not always biologically realistic. Global fit methods allow the evaluation of phylogenetic uncertainty and the contribution of individual host–symbiont links to overall congruence. This may also be important in the light of interpretations regarding an ancestral host (see below).

RESPONSE: *As recommended, we have now implemented PACo and include the results. The analysis is described in the methods section (starting on line 256 of the revised manuscript) and the findings are reported in the results section (starting on line 322 of the revised manuscript). This reviewer raises an important point, that while 75% is a commonly used cutoff in phylogenetic studies, it may not be ideal in phylogenomic studies. However, the lowest bootstrap value retained after collapsing nodes with less than 75% support, was 88%. We have edited the manuscript to indicate which nodes were collapsed at less than 85%, as this is accurate and meets the reviewer's suggested cutoff (see line 287 of the revised manuscript). Yes, we agree that the presence of polytomies could inflate the number of cospeciation events; however, the polytomies represented early divergences in the parasite tree. We predicted and observed that cospeciation was rarer than speciation following host-switching in these early diverging lineages. Therefore, the impact of finding false cospeciation events should not have impacted our interpretation of the results.*

- In Figures S9-S11 the legend does not include an explanation of the node colors and dots vs circles. What is the likelihood for each of these scenarios to occur (Jane)?

RESPONSE: *Thank you for pointing out this oversight in the original manuscript. We have now added text to the figure legends to clarify the meaning of node types and a reference to the software used to generate the figures.*

- Wing lice diversified 15 or 24 mya while doves diversified much earlier (51 or 60 mya). It is hence interesting to estimate where and when the parasites came into play and who the ancestral host/parasite pair could have been. Cophylogeny analysis supports four (!) possible origins on three different continents, however, the basal lineage in the parasite tree indicates that this is likely either Australia or South America. Maybe this could be listed as supportive information. It is unfortunate that other more basal nodes have such low support. Weighed contributions of cophylogenetic signal obtained from PACO could also help to better elucidate the parasites' origin.

RESPONSE: *Unfortunately, cophylogenetic and biogeographic reconstructions can have some level of uncertainty depending on the patterns of diversification. Since PACo does not reconstruct ancestral hosts or events, we don't see a clear path to*

use it to identify the ancestral host or its distribution. We aimed for maximum transparency by documenting the uncertainty that exists, rather than making a claim that is not fully supported by the data.

- Table S1: the numbering system of divergent lineages is according to Boyd et al. [16]. Reference 16 is inexistent in the supplementary data file and is Alavaex et al. 2010 in the main manuscript. I guess you refer to Boyd et al. 2017 [24 in main manuscript]? However, the lineages in this paper are labelled with letters not numbers as in table S1. Please clarify. On another note, please specify methods that led to the results in table headers of tables S1 and S2.

RESPONSE: *Thank you for catching the citation error. We have corrected this issue. The letters in Boyd et al. refer to clades. The numbering system used here follows Boyd et al. (Fig. 1) with regards to species. We have also added the reference to JANE, which was used to specify costs and reconstruct events, which is reported in Table S2.*

- The left section of fig. 2 is a repetition of fig. 1. While this may be necessary to illustrate cophylogeny relationships, I am not a fan of tangled trees and believe that the interpretations on what (host dispersal, host switching, parasite radiation) happened in which order on which continent could nicely be illustrated in an additional figure (map with arrows for introductions and symbols for radiation etc.) and would help the reader to understand the historic course of interaction between different host and parasite species.

RESPONSE: *We agree that it can be difficult for the reader to infer coevolutionary events from a tanglegram. However, these two images convey different information that we believe is critical for the reader to interpret our results. Figure 1 depicts the biogeographic history of doves, which is critical to understanding the roles of vicariance and dispersal by the host doves (which in turn impacted parasite speciation). The tanglegram, figure 2, compares the time-calibrated phylogenies of the host and parasites, demonstrating that the host and parasite phylogenies are frequently discordant and that the doves radiated prior to their parasites. We tried the strategy suggested by the reviewer, but were ultimately unable to convey our results in a concise way required of a figure included in the main manuscript. However, we agree that this information needs to be presented. The information suggested by the reviewer is contained within the supplementary figures (figures S9-S11).*

- Spell out K-Pg boundary at first mention

RESPONSE: *We have done this as suggested on line 387 of the revised manuscript.*

Referee: 2

Comments to the Author(s)
Review of RSPB-2021-1759

This is a really nice manuscript based on timely questions, robust analyses and generally sound interpretations. There are, however, some general issues that, in my opinion, require discussion / clarification, particularly for the non-expert in dove-feather lice associations. Please note the following:

RESPONSE: *Reviewer 2 expressed concerns regarding processes other than cospeciation and host-switching that could impact our results. Our analyses considered these additional processes, and even though they were rare or do not result in parasite speciation, we were careful to limit their impact on our final analysis. Details are given in our individual responses.*

1. The authors consider just two co-evolutionary mechanisms, i.e., cospeciation and host-switching, as drivers of the observed patterns of association of the system under study. I wonder why other processes, e.g., missing-the-boat, duplications, or simple co-accommodation (failure to speciate) were not considered in co-phylogenetic reconstructions. This should be clarified.

RESPONSE: *Duplications, sorting events, and failure to speciate were accommodated in our cophylogenetic reconstructions conducted using Jane. Only one of these processes (duplication) involves speciation by the parasite, which was the focus of our additional analyses. We failed to find evidence for duplication (i.e. speciation of a parasite within a single host species) in our Jane analyses, and indeed it is very rare for a single species of dove to host co-occurring species of wing lice (Columbicola). The paucity of co-occurring parasite species on a single host again suggests duplications are uncommon and supports the niche saturation hypothesis. We have clarified this point in the revised manuscript starting on line 267 and again on 294.*

2. I am not particularly familiar with the degree of specificity of feather lice, but in other systems (e.g., marine helminths and their hosts) there is plenty of new host colonization events without concurrent speciation of the parasite (i.e., co-accommodation). I am aware that co-adaptation of feather lice to new hosts must, in principle, be more stringent, and this might generate speciation as a side-effect. However, this does not necessarily mean that all new host colonizations result in lice speciation, or does it? This should be clarified. More generally, I would like to see a bit of clarification on the assumption (made in this and many other studies of similar nature) that host-parasite associations are univocal (i.e., one parasite species – one host species). If there is even a moderate number of instances in which the same feather louse species occur on more than a dove species, I'd like to know how this phenomenon fits in the relative importance of host-switching vs. cospeciation during radiations.

RESPONSE: *We agree that host-switching is not always associated with speciation by parasites. However, in this case, we explicitly examined speciation in the parasites (i.e. branching points in the parasite tree which are by definition speciation events) and whether they were associated with cophylogenetic processes that can*

result in speciation (i.e. cospeciation, host-switching, duplication). We know of a few cases where a single *Columbicola* species can be found on more than one host species, although most species are host-specific (details have been added to the introduction of the revised manuscript starting on line 62). One such case is *Columbicola tasmaniensis*, which was represented in our dataset and included comparisons of host and parasite trees using JANE (figures S9-S11). This species was excluded in our analysis of cospeciation and host-switching as it represented a failure-to-speciate event in terminal taxa (figures 3, S12). We have noted this in the figure legend of figure S12 in the revised supplementary materials. Furthermore, we described two studies in the original introduction that simulated a host-switch by wing lice in experimental conditions. In these conditions, host-imposed selection led to rapid shifts in host phenotype that also generated reproductive barriers. Therefore, adaptation by a parasite to a new host might often be expected to drive reproductive isolation. The relevant details can be found on lines 64-174 of the revised manuscript.

3. The authors suggest that the saturation of niche space for host-switching is the explanation for the progressive preponderance of cospeciation over time as a mechanism generating diversity. However, this assumption is not really tested and appears to be taken for granted, which is debatable. First of all, the study deals with just 61 (which is, in any event, to be praised!) of the c. 357 spp. of Columbiformes, i.e., we have a partial picture of the history of associations and therefore little idea of how the unknown portion of the history might have influenced patterns of the studied part. More importantly, is there any robust evidence of niche saturation in feather lice? History is by definition idiosyncratic, and a simple phenomenon of pool exhaustion could have resulted in a reduction of host-switching events over time. Besides, the relative importance of cospeciation also depends of the rate of host speciation itself (not to mention other evolutionary phenomena not considered in the analyses –see my first comment). Therefore, it is difficult to make generalizations (or predictions) beforehand. In short, I would recommend the authors to ‘tone down’ their claims on the relative importance of niche saturation and adopt a more pluralistic set of potential explanations (or at least support better their selected explanation).

RESPONSE: *We have toned down the suggestions that our results support the idea of niche saturation, rather we now indicate that the results are consistent with the hypothesis of niche saturation in the introduction (line 200 of the revised manuscript) and discussion (lines 436-450 of the revised manuscript), but could potentially result from other factors, such as greatly increasing rates of diversification in the hosts leading to more cospeciation, or a decline in the availability and abundance of vectors (e.g. hippoboscid flies) for host switching over time.*

Congratulations on a really nice work.

RESPONSE: *Thank you!*

Appendix B

Response to Referee: 2

Comment: The authors have satisfactorily addressed all my queries. I think the paper should now be published.

Response: *We thank you for your helpful comments and are happy that we could address your concerns.*

Response to Referee: 3

Comment: This is a great phylogenomic time-tree of the Columbidae, with a useful biogeographic analysis that shows strong signal of global biogeographic regions. The corresponding wing-louse phylogeny indicates lots of co-speciation. Those results are exciting. A central claim of the paper relates to increasing number of co-speciation events, relative to host-switch events, toward the present. This result is not convincing because it is not appropriately described or tested with respect to a null expectation (see additional details that I describe below). The paper also fails on issues of methods and sample documentation, and the lack of documentation should be easily remedied, but the issue of uncertainty in the louse time tree may be a serious one affecting the interpretation of results.

Response: *We thank the reviewer for the comments, which have helped us clarify some of the points raised by this reviewer. Many of the comments relate to the distinction between extinct diversity and extant diversity. We agree with the reviewer that we cannot address patterns of diversification in now extinct lineages. Indeed, the vast majority of phylogenetic studies cannot address this issue, because they rely on sampling of extant taxa, as we do here. We have now clarified throughout the manuscript that our study relates to patterns of diversification in the extant diversity of parasite taxa and there might be unknown (and unknowable) patterns in extinct lineages that we cannot address. We provide more specific details in the responses below.*

We also note that some of the reviewer's concerns relate to material added in response to comments of Reviewers 1 and 2 from the first round of review (which were both very positive overall). We have now reconsidered some of these additions in light of this additional round of reviewer comments (see below).

We also noted that the reviewer had difficulty finding some of the original data linked to the manuscript, or data used in this study that was associated with a previously published manuscript. We have generated a new data repository that includes both newly generated data and data used in this current study, but generated by a previous study. We also improved the organization of the repository and linked new repository in the revised manuscript draft. At this time, the editor and reviewer can access this new repository using this private link: <https://figshare.com/s/a1314fdb46ffcd3003a>

Comment: The assertion that older dispersal events by doves led to host switching, followed by predominance of younger cospeciation events is central to the paper, the main result reported in the abstract. However, it's problematic in multiple ways. Dove 'dispersal' as the specific driver of ancient inferred host-switch events is a bold assertion because of the great deal of uncertainty about ancient ranges, known range dynamism (i.e. expansion and contraction as alternative to 'dispersal') of many bird

groups through the course of the Cenozoic, and other potential sources of phylogenetic conflict that could cause the appearance of host switching – these caveats should be noted, and assertions made more conservative -- a better fit to the evidence at hand would be to assert ‘apparent host-switch events’ without attributing them to dispersal specifically.

Response: It is not clear that the distinction between dispersal and range expansion is relevant in this context. Certainly, range expansion requires dispersal to a new region. Perhaps the reviewer is attempting to distinguish dispersal that leads to speciation and dispersal that does not lead to speciation (range expansion)? However, either could facilitate host-switching events by parasites. In either case, the method that we employ to reconstruct biogeographic history (BioGeoBears) can accommodate a wide variety of biogeographic models, including making the distinction between dispersal leading to speciation and dispersal not leading to speciation. In the context of more ancestral nodes, in which more uncertainty in ancestral range reconstruction and ancient host switching exists, we have now added additional caveats regarding this uncertainty. In the case of the four examples of more recent dispersal events leading to host-switching, these are all examples in which the dispersing taxa is deeply embedded within lineages with an alternate biogeographic distribution and in which the direction of host-switching is stable across all alternative reconstructions, given the pattern of the phylogenetic tree. Thus, we are generally confident that these latter cases provide clear examples of the processes we describe.

Comment: Methods section: where is the description of the louse molecular data and tree? Except the note about 977 loci in the caption of Fig. 2, I couldn’t find it (another tiny note suggesting it came from Dryad?). The time-dependence of many of the results means that more attention needs to be given to the louse fossil calibrations and the uncertainty associated with them. The description given indicates that there are mostly secondary calibrations involved – the basis for these needs to be described. What assumptions were made, based on what prior data and analyses, and where is the degree of uncertainty described? (this reviewer is satisfied with the avian timetree because of the substantial previous work on that, and general concordance with fossil record). Uncertainty of node timing is not depicted anywhere – but it should be included with timetrees by default – to not do so is unhelpful at best. Table S1 is inadequate because it neither contains useful information about the basis for the parasite calibrations, nor information about how the calibration point was used (max? min? what prior distributions?).

Response: Louse Tree: Unlike the dove tree, the louse molecular tree and data results were from a prior publication (Systematic Biology 2017; 56: 656-672), which we had cited in the Methods section and an active link to the parasite tree in DRYAD was provided in the original Data Accessibility Statement. Our apologies if this was insufficient. To address this comment, we have added details to the Methods section, starting on lines 134 and 157, to fully clarify the source of the louse tree, and to describe which of trees associated with the 2017 paper were used here. Likewise, we have now restated much of the methods used to generate the louse tree in the supplementary methods (starting on lines 148, 165, 181, and 197). We believe this change will allow the reader to more easily and quickly understand the approach for inferring both host

and parasite phylogenies. Additionally, we have copied the parasite data into the new data repository associated with this publication. Each element of the data repository is clearly labeled as being data from lice or doves. Louse calibrations/Table S1: Our apologies if the calibrations used were not clear. The confusion seems to be due to the use of the term “variable” in table S1. To understand the uncertainty of louse divergence times, we felt that it was best to replicate the analysis using all possible combinations of root and internal calibrations. Therefore, in table S1 we chose to use the term variable. As was described in the original manuscript (now on line 169 of the revised manuscript), internal calibrations, those denoted with “^variable” in table S1, are linked to min-max calibration ages based on terminal host-parasite cospeciation events, given the 95% CI for host speciation time. The min-max values are provided in figure S5-S8, along with the associated root age. We have added details the header and footer of table S1 that reinforce the link between table S1 and figures S5-S8 regarding the min-max calibrations. We also linked those figures with text in the main manuscript (lines 163, 171, and 174), where we had previously only linked in table S1. Node uncertainty: As with any study of this size, we had to make difficult decisions about what information to include or exclude in the figures. The 95% confidence intervals were left off the figures S3-S8 as they made the figures cluttered and difficult for the reader to readily identify our calibrations times (which were included in these figures). The 95% CI were included in all time-trees deposited in the supplementary data. To address this comment, we have now added six supplementary figures (figures S9-S14) that display the 95% CI. Overall, 95% CI suggested greater uncertainty in the dove tree; however, as this reviewer noted the dove tree agrees with previous findings. Uncertainty in the louse tree was lower. Important to this study, the 95% CI surrounding the initial split in the dove tree does not overlap with that of the 95% CI surrounding the initial split in wing lice in any of the four possible comparisons.

Comment: Line 260: “The median age of louse speciation events due to cospeciation was significantly younger than those due to host-switching (Wilcoxon Signed Rank test $P = 0.021$).” This weak statistical result is carrying a lot of weight in this paper (central result described in abstract), yet I assert that it is invalid for a simple reason: each host-switch event can erase the signal of cospeciation events that preceded it; therefore, ancient cospeciation events are more likely to have been erased by subsequent evolution; therefore, a null model under constant rates of cospeciation and host switching over time is likely to lead to the appearance of increased rates of cospeciation toward the present.

Response: We agree with the reviewer that we cannot address patterns in lineages that are now **extinct**. When we are mapping out the timing of cospeciation and host-switching, we are doing this over the speciation events that produced the **extant** diversity of wing lice (i.e. the speciation events in the phylogenetic tree of lice). Almost no phylogenetic studies (let alone cophylogenetic studies) incorporate patterns of extinct diversity, given the extreme difficulties in doing so (i.e. a nearly complete fossil record).

However, we disagree with the premise of the reviewer that host-switching somehow erases earlier cospeciation events in lineages leading to extant taxa. As an example, take the host phylogeny (A,B),(C,D). Let's say the parasite cospeciates at all three of these

host speciation events. Then the parasite lineage on host ancestor C switches (with speciation) to host B, resulting in the parasite tree (A,B1),((C,B2)),D). This does not erase any signal of the ancestral cospeciation event between the (A,B) versus (C,D) clades. Even if the original parasite B1 went extinct, the signal from the original cospeciation event would still be there in the parasite phylogeny and be properly reconstructed by the Jane cophylogenetic reconstruction algorithm. Only if the parasite on A went extinct would this signal be lost. But in that case, the original cospeciation event now has nothing to do with the **extant** diversity of parasites, because they now originate on the ancestor of the (C,D) clade and diversify largely by host-switching **after** the diversification of their hosts. This latter case is exactly analogous to the situation we see in *Columbicola*, in which the timing of parasite diversification occurs largely after the diversification of their hosts. In fact, this timing difference alone (even without formal cophylogenetic reconstruction analysis) provides strong implication that at least some of the early diversification of the parasites occurred via host-switching. To address this comment, we have now clarified throughout the manuscript that our study relates to the **extant** diversity of the parasites (including changes to lines 47, 249, 282, 321, 322, 324, 340, and 347).

Comment: Lines 263-265: The results are said to be “incompatible” with the time trees, but was error properly carried and reported for the louse time tree? How is ‘incompatibility’ quantified, and what is the degree of ‘incompatibility’?

Response: The mean node age and 95% CI for the earliest divergence within the dove and wing louse trees do not overlap in any of the four possible comparisons. We have now added clarification regarding the lack of overlap of the 95% confidence intervals for the relevant nodes to clarify this point starting on line 227 of the revised manuscript (see also response above regarding node uncertainty).

Comment: Line 286: “as parasite-free host species declined in frequency” That’s a bold inference to make — can’t host switching happen even by displacing a current louse occupant? What proportion of extant dove species are ‘parasite-free’? If there’s no evidence for this specific claim, the following claim also is an overreach: “These results indicate that hosts without parasites may have provided new ecological opportunities for parasite adaptive radiation.”. Alternatively, ancient host-switches reflect the outcome of competition...

Response: We agree with the reviewer that we cannot make any strong inference about possible extinct taxa from a study of extant lineages. Although, we feel that the existence of parasite free dove lineages might provide a plausible scenario for the role that host-switching played in their parasitic lice, we agree that we cannot necessarily infer that is the case. We have changed the wording as suggested by the reviewer (lines 283-288 of the revised manuscript). We also now point out in more detail that experimental work focusing on host-switching, with and without potentially competing taxa, might help clarify the plausibility of these two scenarios. Indeed, our results provide additional motivation for conducting such experiments, as a combination of cophylogenetic pattern and experimental data can provide mutually illuminating insights into the patterns and process of parasite diversification (changes made on lines 289-291 of the revised manuscript).

Comment: Line 333-4: “It appears that a single wing louse lineage was isolated within a single geographical region prior to diversification.” Appearances are deceiving... Just as coalescence of a gene tree to a single individual in the past does not imply a population size of one... other lice may have been on other doves, they may have simply failed to leave extant descendants! In other words, I don’t think it’s OK to make an (arguably) implausible ecological hypothesis simply because of an absence of evidence. It appears that all or nearly all of these 61 columbid species are ‘infested’, therefore it’s parsimonious to suggest that all columbid species have always been ‘infested’, with periodic extinctions driven by host-switching.

Response: Again we agree with the reviewer that we are unable to make inferences about extinct parasite lineages. We have now modified this text to clarify this issue, leaving open the possibility that these ancestral dove lineages may have had lice that are now extinct and completely replaced by the extant wing louse diversity.

Comment: Line 348-9: “However, other processes could conceivably have resulted in this pattern. For example, an increase in the rate of host diversification could have provided additional opportunities for cospeciation, or the opportunity for host-switching could have diminished over time.” This implies that the method by which events were quantified did not correct for ‘opportunity’. If this is the case we should always expect increasing co-speciation toward the present, by default. The authors should perhaps return to the white board to reconsider the appropriate way to identify this supposed pattern.

Response: We had added this somewhat speculative suggestion in response to comments of Reviewer 2. We recognize that the patterns we have found could conceivably have more than one explanation. Thus, we further modify the text to indicate that a niche saturation hypothesis is consistent with the data, but not necessarily strongly supported. We removed the more speculative explanation regarding changes in diversification rates, for which we have no evidence.

Comment: Line 350-1: “(e.g. by a decline in the abundance of hippoboscids, which can vector wing lice between host species [48])” There is no cited evidence here for a such a decline in hippoboscids (I looked up and read reference 48 to check), making this entirely speculative – please delete it.

Response: Again, this highly speculative idea was added in response to the comments of Reviewer 2. In light of the highly speculative nature of this hypothesis, and the complete lack of evidence, we have now removed it.

Comment: The sample table is buried deeply in the Dryad accession among many small files that document minor components of the analyses. It is also deeply inadequate, with hodge-podge sample identifiers and formats (use Darwin Core format where possible, and/or at least identify the identifier type, whether personal catalog number or tissue collection number, bird collection number, etc.), many blank cells, and institutions of origin not clearly indicated. Catalog numbers from IL natural history survey attributed to samples loaned from other collections are also present – I doubt that curators of those collections would be OK with these additional ascribed identifiers, but the authors here at least have the obligation to provide clarification. I

further strongly suggest that this table be included (or repeated) in the supplement of the present paper, if not the main paper itself. Finally, where is the sample table for the lice?... same principles of documentation apply.

Response: The details of louse sampling were part of a prior publication describing the data, methods, and results for the louse phylogeny, *Systematic Biology* (2017; 56: 656-672) and was also accessible through the active link to DRYAD provided in the data accessibility statement. Thus, some of the details regarding the louse sampling was not included in the present manuscript. However, to address this comment and simplify data accessibility we have now copied the louse data collection identifiers into associated repository for this paper. The Illinois Natural History Survey (a museum) maintains a cataloged collection of tissues samples and that collection was the source for the material for this study. The INHS identifiers represent the primary catalog numbers for specimens used in this study. These identifiers were often tied to another identifier, the deposition of the avian skin, which are in other institutions. Those identifiers were also included.

Comment: The discussion of the lice as an ‘adaptive radiation’ is a confusing and unnecessary to this paper.

Response: Villa et al. (2019; *Proc Natl Acad Sci USA* 100:15694-15699) and Bush et al. (2019; *Am Nat* 176:529-535) provided direct experimental evidence of adaptative trait evolution in wing lice, that were linked to reproductive traits, directly following a host-switch. Our current study builds on these results by exploring broader patterns in the biogeographic and cophylogenetic context for an adaptive radiation, and thus we feel it is an appropriate conceptual framework for our study. As wing lice are emerging as an experimental system to examine adaptive speciation, we believe the link between previous work and the findings presented here will be an important and foundational contribution, guiding future work integrating both experimental and cophylogenetic studies. Furthermore, the two prior reviewers on the first submission of this manuscript felt that “The paper is well written and interesting to a broad range of readers from parasitology, ecology, evolutionary biology and general biology” (Reviewer 1); and “This is a really nice manuscript based on timely questions, robust analyses and generally sound interpretations. Congratulations on a really nice work.” (Reviewer 2). Thus, we feel confident in our framing of the overall background for the paper.

Comment: Biogeobears analysis is poorly documented – many very impactful parameter and model settings are not considered or described.

Response: Thank you for pointing out this oversight. We have now added a section to the supplementary methods detailing our usage and approach to biogeographic inference using BioGeoBEARS. These changes begin on line 215 and end on line 223 of the supplementary methods.